# The Ratio of GrzB^+^ − FoxP3^+^ over CD3^+^ T Cells as a Potential Predictor of Response to Nivolumab in Patients with Metastatic Melanoma

**DOI:** 10.3390/cancers13102325

**Published:** 2021-05-12

**Authors:** Giosuè Scognamiglio, Mariaelena Capone, Francesco Sabbatino, Annabella Di Mauro, Monica Cantile, Margherita Cerrone, Gabriele Madonna, Antonio Maria Grimaldi, Domenico Mallardo, Marco Palla, Sabrina Sarno, Anna Maria Anniciello, Maurizio Di Bonito, Paolo Antonio Ascierto, Gerardo Botti

**Affiliations:** 1Pathology Unit, Istituto Nazionale Tumori IRCCS Fondazione “G. Pascale”, 80131 Napoli, Italy; giosue.scognamiglio@istitutotumori.na.it (G.S.); annabella.dimauro@istitutotumori.na.it (A.D.M.); m.cantile@istitutotumori.na.it (M.C.); margherita.cerrone@istitutotumori.na.it (M.C.); a.anniciello@istitutotumori.na.it (A.M.A.); m.dibonito@istitutotumori.na.it (M.D.B.); 2Melanoma, Cancer Immunotherapy and Development Therapeutics Unit, Istituto Nazionale Tumori IRCCS Fondazione “G. Pascale”, 80131 Napoli, Italy; g.madonna@istitutotumori.na.it (G.M.); a.grimaldi@istitutotumori.na.it (A.M.G.); d.mallardo@istitutotumori.na.it (D.M.); m.palla@istitutotumori.na.it (M.P.); p.ascierto@istitutotumori.na.it (P.A.A.); 3Oncology Unit, AOU San Giovanni di Dio e Ruggi d’Aragona, 84125 Salerno, Italy; fsabbatino@unisa.it; 4Department of Experimental Medicine, University of Campania “Luigi Vanvitelli”, 80138 Napoli, Italy; sabrina.sarno@unicampania.it; 5Scientific Direction, Istituto Nazionale Tumori IRCCS Fondazione “G. Pascale”, 80131 Napoli, Italy; g.botti@istitutotumori.na.it

**Keywords:** melanoma, immunotherapy, tumor microenvironment, multiplex immunostaining, immunoscore

## Abstract

**Simple Summary:**

Despite the recent success of immunotherapy in the treatment of malignant melanoma, many patients still do not benefit from these treatments, due to their failure to activate an antitumor immune response them. There is therefore a need to select patients who can truly benefit from these treatments. We have focused our study on immune cells present in the tumor microenvironment, and we have developed a formula (ratio) that correlates with the response to anti-PD1 therapy and progression-free and overall survival, based on the numerical difference between GRZB^+^ and FOXP3^+^ cells over the total CD3^+^ lymphocytes. This developed ratio could be useful to better select patients that may or may not benefit from anti-PD-1 treatment.

**Abstract:**

The understanding of the molecular pathways involved in the dynamic modulation of the tumor microenvironment (TME) has led to the development of innovative treatments for advanced melanoma, including immune checkpoint blockade therapies. These approaches have revolutionized the treatment of melanoma, but are not effective in all patients, resulting in responder and non-responder populations. Physical interactions among immune cells, tumor cells and all the other components of the TME (i.e., cancer-associated fibroblasts, keratinocytes, adipocytes, extracellular matrix, etc.) are essential for effective antitumor immunotherapy, suggesting the need to define an immune score model which can help to predict an efficient immunotherapeutic response. In this study, we performed a multiplex immunostaining of CD3, FOXP3 and GRZB on both primary and unmatched in-transit metastatic melanoma lesions and defined a novel ratio between different lymphocyte subpopulations, demonstrating its potential prognostic role for cancer immunotherapy. The application of the suggested ratio can be useful for the stratification of melanoma patients that may or may not benefit from anti-PD-1 treatment.

## 1. Introduction

Melanoma is the most serious form of skin cancer and is responsible for most skin cancer-related deaths. The incidence of malignant melanoma has dramatically increased over the years [1] but, fortunately, there has also been considerable progress in its treatment, including the use of targeted and immune checkpoint blockade therapies. In particular, major advances have been achieved in targeting the immune evasion phase of tumors using drugs blocking the inhibitory control points that regulate the immune system, such as programmed death-1 (PD-1) and cytotoxic T-lymphocyte antigen-4 (CTLA-4) [2]. In fact, both anti-PD-1 (nivolumab, pembrolizumab) and anti-CTLA-4 (ipilimumab) agents have demonstrated unprecedented substantial benefit and durable responses in patients with metastatic melanoma, regardless of BRAF V600 mutation status [3]. These approaches have revolutionized the treatment of melanoma, but are not effective in all patients, resulting in responder and non-responder populations. Inherent resistance of melanoma cells and the host immune response of the tumor immune microenvironment (TIME) represent major limitations for the treatment of melanoma [4]. Melanoma is often infiltrated with immune cells and pre-existing tumor immune cell infiltration is considered to be an important factor determining successful immune checkpoint inhibition and treatment response. The extent of tumor-infiltrating lymphocytes (TILs) has been shown to be an independent predictor of survival irrespective of the treatment type [5,6]. Lymphocytic infiltrate regulates the tumor microenvironment (TME) through cytotoxic and immunosuppressive mechanisms. Regulatory T cells (Treg) are an immunosuppressive population of T cells, able to provide peripheral immune tolerance by secreting immunosuppressive cytokines and suppressing the activation and proliferation of cytotoxic T cells. They contribute to the induction of immune tolerance to melanoma cells and are characterized by a stable expression of the transcription factor FOXP3 [7]. Moreover, an increased number of Tregs in the TME has been shown to correlate with a poor prognosis in various types of malignancies, including melanoma [8]. On the other hand, both cytotoxic T lymphocytes (CTL) and natural killer (NK) cells, which play a direct role in the antitumor activity, lyse target cells through the exocytosis of granules containing perforin and granzymes. Granzyme B (GRZB) is the most abundant of these and has a well-characterized intracellular role in the targeted destruction of compromised cells by CTLs [9]. Favorable clinical outcomes are strongly associated with the presence of GRZB^+^ cells in melanoma patients [10]. In the context of anti-PD-1 therapy for melanoma, CD8^+^ T cell density at the invasive tumor edge has been correlated with response to anti-PD-1 treatment [11]. Both CD8^+^ and CD4^+^ T cells represent the most prevalent immune-infiltrating populations found nearby melanoma cells, but recent studies revealed that other types of immune cells, such as M1 or M2 macrophages, also correlate with melanoma prognosis [12]. Recently, an immunoscore model in melanoma has been established in order to predict antitumor response. Among the features of the TME, the immune cell phenotype efficiently predicted the response to anti-PD-1 therapy [13]. Furthermore, studies of gene expression profiling showed that melanoma patients treated with immune checkpoint inhibitors showed better treatment efficacy if they expressed genes related to the inflammatory response. These gene profiles included markers for CTLs (e.g., CD8A, GRZB, perforin 1, Th1 cytokines or chemokines, MHC class II) and interferon (IFN)-γ gene signature [14]. Given the immunogenic nature of melanoma [15], as well as the poor prognosis associated with metastatic disease, we used a multiplex immunostaining approach to evaluate the immune cell infiltration in both primary and unmatched in-transit metastatic melanoma lesions of patients retrieved before treatment with the anti-PD-1 nivolumab. The discovery that favorable immune contexture is linked to prolonged patient survival—and, more specifically, that intratumoral CTLs have powerful prognostic value—provided the foundations for the development of the immunoscore, allowing the quantification of two T cell subsets (CD3 and CD8) in two tumor regions (core and invasive margin of tumors). Galon et al. [16] demonstrated that the immunoscore had superior prognostic value compared to the traditional TNM system, therefore validating the immunoscore as the first immune-based scoring system. Thus, we evaluated the presence of such immunological markers both in intratumoral and peritumoral areas, proposing a simple lymphocyte ratio that could be useful for the stratification of melanoma patients that may or may not benefit from anti-PD-1 treatment.

## 2. Materials and Methods

### 2.1. Tumor Samples

From the biobank of the Istituto Nazionale Tumori Fondazione “G. Pascale” (Naples, Italy), we recovered excised melanoma tumor samples from thirty patients (16 males, 14 females, median age 66.5 years) subsequently treated with nivolumab as monotherapy from September 2014 to November 2017. Nivolumab was administrated at the dosage of 3 mg/kg every 2 weeks until disease progression or unacceptable toxicity appeared. In particular, we retrospectively recovered 14 primary melanomas (46.7%) and 16 unmatched in-transit melanoma metastases (53.3%), i.e., the primary and melanoma metastases were not from the same patient. All tumor samples were excised from melanoma patients before treatment with anti-PD-1. All patients provided signed informed consent and the institutional review board (IRB) approved the study protocol. The presence of tumor cells in formalin-fixed paraffin-embedded (FFPE) tissues was reviewed by two expert pathologists (GB and AA) according to American Joint Committee on Cancer (AJCC) classification criteria. Medical records for all patients were retrieved for clinical information. The following clinical and pathological parameters were evaluated: patient age at start of treatment, histological features, tumor differentiation, line of treatment, BRAF status, lactate dehydrogenase (LDH) level and clinical response (Table 1).

### 2.2. Immunohistochemistry (IHC)

FFPE tumor tissue sections of 4 μm thickness were cut onto adhesive slides. For each sample, the following markers were evaluated: CD3, CD8, FOXP3, GRZB, MMR panel (MLH1, MSH2, MSH6 and PMS2) and PD-L1 (Appendix A).

For the MMR panel, CD3 and CD8 immunohistochemistry, consecutive slides were processed using the automated immunostainers, while, for FOXP3 and GRZB, manual immunostaining was performed. PD-L1 expression was detected by an E1L3N clone for one hour after epitope retrieval in a pressure cooker at 110 °C. Secondary antibodies and manufacturers and incubation conditions for primary antibodies are summarized in Appendix A.

Subsequently, to quantify the number of CD3, GRZB and FOXP3-stained cells on the same slide, we performed triple staining using a sequential multiplex immunostaining approach for each tissue sample [17]. In detail, the slides were incubated for 12 min (min) at 110 °C in Cell Conditioning Solution 1 (Ventana, Roche, Oro Valley, AZ, USA), using a commercial steamer as the heat source (Biocare Medical, Pacheco, CA, USA, Decloaking Chamber DC12). After cooling for 20 min, slides underwent a 5 min incubation with a peroxidase-blocking reagent and 10 min incubation with a protein serum block (1% goat serum, 5% BSA in PBS). After pretreatment, the slides were incubated with the rabbit anti-FOXP3 antibody for 60 min at room temperature. The goat anti-rabbit plus HRP visualization reagent (Agilent Technologies Inc., Santa Clara, CA, USA) was incubated for 40 min for FOXP3-specific antibody detection. Brown color was developed using 3,3-diaminobenzidine tetrahydrochloride (DAB; Agilent Technologies Inc.). After incubation with a Denaturing Solution (Biocare Medical) for 5 min, the rabbit anti-GRZB was incubated for 60 min at room temperature. The Donkey Anti-Rabbit IgG H&L (Alkaline Phosphatase) preadsorbed (ab98496, Abcam, Cambridge, UK) was incubated for 60 min at room temperature and Vulcan Fast Red Chromogen Kit 2 (Biocare Medical) was used as a second chromogen for 15 min. Following incubation with a denaturing solution for 5 min, the third antibody, the rabbit anti-CD3, was incubated for 60 min at room temperature. The goat anti-rabbit plus HRP visualization reagent (Agilent Technologies Inc.) was incubated for 40 min. Finally, the slides were incubated with the Vina Green™ Chromogen Kit (Biocare Medical) chromogen for 5 min.

The expression of CD3 (membrane and cytoplasmic staining), CD8 (membrane staining), FOXP3 (nuclear staining), and GRZB (cytoplasmic staining) was evaluated within the intratumoral and peritumoral areas. The lymphocyte count was established based on the number of positive cells, counted by two different pathologists and, where possible, in 20X fields. The membrane expression of PD-L1 was evaluated only on tumor cells.

### 2.3. Statistical Analysis

Correlation of biomarker expression was analyzed by Spearman’s rank correlation coefficient. Non-parametric tests were used to compare independent groups of numerical data. Differences in the expression of CD3, CD8, GRZB, FOXP3 and PD-L1 according to gender, primary/metastatic tumor, BRAF status and MMR status groups were analyzed using the Mann–Whitney U-test. In order to correlate the biomarker expression, the Spearman’s rank correlation coefficient was evaluated. Differences in the expression of CD3, CD8, GRZB, FOXP3 and PD-L1 according to the best overall response rate (BORR) were analyzed using the Kruskal–Wallis rank test. Survival curves were estimated using the Kaplan–Meier method and differences among groups were analyzed using the log-rank test. Progression-free survival (PFS) was defined as the time between the beginning of therapy and the appearance of signs of disease progression. Data on survivors were censored at the last follow-up. *p* < 0.05 was considered statistically significant. All tests used were two-tailed. All statistical analyses were carried out using SPSS version 20.0 software (SPSS, Chicago, IL, USA).

## 3. Results

### 3.1. Clinicopathological Features of Melanoma

Thirty specimens from melanoma patients treated with nivolumab were included: 14 from primary melanomas (46.7%) and 16 from unmatched in-transit melanoma metastases (53.3%). The age range of patients was 26–77 years, with a median of 66.5 years. Fourteen of 30 (46.7%) patients were female. BRAF (V600) mutations were detected in nine patients (30%). Baseline serum LDH values were elevated in six patients (20%), normal in 21 patients (70%) and unavailable for three patients (10%). Best overall response rate (BORR) to nivolumab was 33% (10/33 pts), with a complete response (CR) in 2/30 patients (6.6%), partial response (PR) in 8/30 patients (26.7%), and stable disease (SD) in 9/30 patients (30%), while 11/30 patients had disease progression (PD) (36.7%). Median OS for these patients was 20 months and median PFS was 16 months; median time of follow-up for them was 20 months (range 1–54 months). The details of the clinicopathological data are shown in Table 1. Regarding the main tumor characteristics of primary melanomas (*n* = 14), tumor thickness ranged from 0.2 to 12 mm and ulceration was present in 10/14 (71.4%) of the tumors analyzed. Two melanomas (14.3%) were pT1, two (14.3%) were pT2, six (42.9%) were pT3 and four (28.6%) were pT4 (Table 2).

### 3.2. Quantification of Single TIL Biomarkers and PD-L1 in Primary Tumors and Unmatched In-Transit Metastatic Melanoma Tissues and Its Correlation with the Pathological Features

Lymphocyte biomarkers were analyzed to determine whether the number of lymphocyte subpopulations had a predictive or prognostic role in melanoma patients treated with immunotherapy. CD3 and CD8 cells were present in the peritumoral areas of all analyzed samples. The median number of CD3^+^ lymphocytes was 143 (range 15–548) and median number of CD8^+^ cells was 51 (range 5–343). The median number of GRZB^+^ cells was 5 (range 0–32), while the median number of FOXP3 cells was 9 (range 0–150). GRZB^+^ cells were absent in three cases and FOXP3^+^ cells in two cases (Figure 1).

PD-L1^+^ cancer cells showed membrane immune reactivity (Figure 1f). PD-L1 expression was observed in 10/30 cases (33.3%) (range of PD-L1 expression 3–20%), of which 3/14 (21.4%) were primary melanomas and 7/16 (43.7%) were unmatched in-transit melanoma metastases.

The correlations between biomarker expression and available clinicopathological features (gender, primary vs. metastases, BRAF status and BORR) are summarized in Table 3.

### 3.3. Correlation between Biomarkers

CD3 were positively correlated with CD8 (*p* < 0.001) and GRZB (*p* = 0.009). CD8 T cells were directly correlated to FOXP3 (*p* = 0.002) and GRZB (*p* = 0.038) T-cells. Correlations between the expressions of single markers with each other are shown in Table 4.

### 3.4. Simultaneous Evaluation of CD3, FOXP3 and GRZB through Triple Immunostaining

In order to verify if the peritumoral immune contexture was more cytotoxic or more immune-suppressive and thus evaluate the balance between cytotoxic and immune suppressive activity in the melanoma microenvironment, we simultaneously quantified the number of CD3, GRZB and FOXP3-stained cells by a multiplex staining approach on all analyzed cases (Figure 2).

We applied a formula considering the numerical difference between the GRZB^+^ cells (cytotoxic cells) and the FOXP3^+^ cells (immunosuppressive cells), on total CD3^+^ cells. The formula is schematized as:Ratio = (# of GRZB − # of FOXP3)/# of CD3.

A negative ratio result meant an immunosuppressive environment, while a positive ratio was indicative of an active cytotoxic environment (range of ratio results: −0.38 to 0.16).

We did not find any statistical correlation between the multiplex staining, the clinicopathological features and BRAF mutational status. However, we found a relationship between the applied ratio and BORR (*p* = 0.025) (Figure 3, Table 5).

The positive value obtained by the ratio of lymphocyte subpopulations was mainly present among those who responded to treatment versus those who did not respond.

### 3.5. Survival Analysis

Based on the distribution of cases, a cut-off based on the median ratio was used to stratify melanoma patients to determine its prognostic significance. Kaplan–Meier PFS and overall survival (OS) curves showed a trend between the values of ratio (cut-off: −0.05) and the duration of response: patients with a low ratio (<−0.05) had lower PFS and OS compared with patients with a higher ratio (≥−0.05) (*p* = 0.022 and *p* = 0.016, respectively) (Figure 4).

## 4. Discussion

The treatment and prognosis of melanoma, particularly in the metastatic setting, has changed significantly over the past decade. Based on the knowledge of the melanoma biology and its immunogenicity, new therapeutic strategies have been developed with antibodies directed to specific targets, including anti-PD-1 and anti-CTLA-4 [18,19]. Some clinical trials have evaluated the efficacy of anti-PD1 treatments in melanoma, especially in combination with ipilimumab, which leads to a significantly greater clinical benefit compared to ipilimumab alone [20].

These clinical trials showed clearly superior median and long-term survival of anti-PD-1-based therapy as compared to any historical treatment options for metastatic melanoma. However, a significant number of patients do not respond to immune checkpoint blockade. Currently, there are no clinically available biomarkers to predict responding patients in melanoma [21]. Anti-PD-1 therapy unleashes the pre-existing antitumor immune response [22]; therefore, biomarkers that represent pre-existing tumor immune phenotypes could be used to predict the response to anti-PD-1. Furthermore, interactions between malignant and immune cells compose the TME, which plays a fundamental role in dampening or enhancing the immune response. The knowledge of the effects of immunotherapy on the TME is therefore crucial to reveal the mechanisms of action of these drugs, to increase the efficacy of current molecules, and to help to select those patients that will benefit most from treatment.

The IHC multiplex approach represents a simple tool for the in situ determination of specific markers of the TME and can be widely used in histopathological diagnostic laboratories. Furthermore, it has already been shown that multiplex IHC is a powerful investigative tool and can be used to assess the immune context of metastatic melanoma [23].

In this study, we analyzed the potential prognostic and/or predictive role of several TIL markers (CD3, CD8, FOXP3, GRZB and PD-L1) in primary melanoma and unmatched cutaneous metastases. We wanted to consider the key players of adaptive immune responses based on T cells because the activation and expansion of these cell types is necessary for long-term melanoma remission and for response to treatment. Furthermore, the intratumoral infiltration of activated lymphocytes derives from the other immune cells yet residing in the tumor niche, such as macrophages, neutrophils and dendritic cells that phagocytize dead melanoma cells and present cancer antigens activating secondary adaptive immune responses.

Among all available patient samples, we selected exclusively those excised in proximity to the start of nivolumab therapy. In total, 12 patients were treated in the first line with nivolumab, while 18 patients were treated in the second, third or even fourth line with it. We know that both immunotherapy and BRAF-targeted agents may affect the presence of immune cells and their functionality [24,25].

The analysis of single markers highlighted the presence of CD3 and CD8 cells in the peritumoral areas of all analyzed cases, whereas GRZB and FOXP3 showed heterogeneous expression. PD-L1 was expressed on tumor cells in around 20% of primary melanoma and in around 43% of unmatched in-transit melanoma metastases. Finally, in all samples analyzed by single staining, no statistically significant correlation was found between the number of CD3^+^, CD8^+^, FOXP3^+^, GRBZ^+^ immune cells or PD-L1 expression on tumor cells and the clinicopathological features, including BRAF mutational status. The low number of samples and their potential heterogeneity are possible reasons that influenced these results, which seem discordant with those of the work of Tumeh et al. [11]. In the latter, the density of CD8 T cells in the invasive margins of the pretreatment melanoma samples appeared as a good predictive parameter for response to anti-PD-1 treatment. However, to date, it is still unclear which characteristics of infiltrating CD8 T cells are crucial for tumor control and whether they are influenced by immunotherapy. Furthermore, limited data exist comparing PD-L1 status in primary versus metastatic melanoma lesions, with few matched sets available from the same patient. FFPE primary melanoma samples may be most readily available, but they may not reflect the overall immunologic state that currently exists in a given patient and may not capture the beneficial effect that therapy is having on other sites of disease that are dependent on PD-L1 signaling. There is currently great interest in understanding mechanisms that drive the expression of PD-L1 in the TME, since it can be expressed by tumor, immune and endothelial cells and its expression can function locally to dampen antitumor immunity. While IFN-g is generally thought to be the primary T cell-derived cytokine responsible for adaptive PD-L1 expression, a recent study [26] described several additional TME-resident cytokines, including IL-10 and IL-32g, which are capable of promoting PD-L1 expression on monocytes but not on tumor cells, becoming an important source of immunosuppression in the TME.

The limitation of single-stain IHC has already been demonstrated in other studies. A single marker may be expressed on more than one cellular population. For example, CD8 can represent both T cells and NK cells, while FOXP3 alone can represent T effector and Treg cells [27]. On the contrary, the simultaneous evaluation of several TME biomarkers has already been shown to have a more adequate prognostic or predictive value.

Previously, we conducted one of the first immunoscore studies in melanoma using a multiplex IHC approach, in order to evaluate the expression of CD3, CD8, CD20 and FOXP3 in metastatic lymph nodes [28]. Subsequently, we conducted a study of 200 FFPE samples from metastatic melanoma patients treated with ipilimumab in which different immune populations were assessed using a digital image analysis application to characterize the immune infiltrate expression of CD3, CD8, CD20, FOXP3, CD163 and PD-L1, in order to investigate the predictive power of the immunoscore. In this study, we found an association between clinical benefit from ipilimumab therapy and the coexistence of low densities of CD8^+^ and high densities of CD163^+^ PD-L1^+^ cells at the periphery of the tumor [12]. Recently, Sharma et al. analyzed TILs from paired pre- and post-ipilimumab-treated melanoma tissues for the expression of multiple TIL markers and showed that ipilimumab increased the frequency of CD8^+^ and CD4^+^ T cells but not FOXP3^+^ Treg cells [29].

In this study, a multiplex sequential immunostaining of CD3, FOXP3 and GRZB on all analyzed cases was performed, establishing a ratio between different lymphocyte populations. We have introduced a new ratio to evaluate if the peritumoral regions were more cytotoxic or immune-suppressive. The ratio is obtained by the difference between GRZB^+^ cells and FOXP3^+^ cells to total CD3^+^ T cells. This ratio did not show statistically significant associations with clinicopathological features or BRAF mutational status, but it was correlated with BORR (*p* = 0.025). In fact, a positive value obtained by the ratio of lymphocyte subpopulations was mainly present among patients who responded to the treatment versus those who did not respond. Moreover, Kaplan–Meier analysis of PFS and OS showed a trend between the value of ratio (≥−0.05) and the duration of response, with higher ratios associated with a longer PFS and OS (*p* = 0.022 and *p* = 0.016, respectively).

These results were obtained analyzing samples from a small number of patients and must be confirmed in future studies with larger cohorts. We considered both primary melanomas and in-transit metastases, both located in cutaneous tissues. For this reason, we decided to exclude other types of metastasis, such as lymph node and visceral ones, which certainly have a very different tumor microenvironment. Moreover, this choice was also related to the ready availability of these pretreatment samples. Despite these limitations, our study suggests that the simultaneous quantification by multiplex IHC of CD3, FOXP3 and GRZB-positive T cells in the same tissue area, and the application of the suggested ratio, can be a useful tool for the stratification of melanoma patients that may or may not benefit from anti-PD-1 treatment.

For optimal prognostic-predictive stratification of patients, a series of panels to evaluate simultaneously the immunoscore of the TME markers, analyzing the expression profiles of more than 700 immune-related genes [13], have been performed. However, these approaches are very expensive, and not yet fully standardized, and they require experience and bioinformatics expertise. Finally, an integrated approach of mRNA/protein quantification may be the best way to define the complexity of the TME.

## 5. Conclusions

In this study, we have introduced a new predictive factor of responsiveness to anti-PD1 treatment of melanoma patients based on a simple multiplex IHC analysis measuring the difference between the number of cells expressing the cytotoxic cell marker GRZB and the number of cells expressing the regulatory T cell marker FOXP3 divided by the total number of CD3^+^ cells in the peritumoral region. Although the number of samples analyzed was small, we found that higher ratios were positively correlated with BORR and with PFS and OS, indicating that this could be a useful measure in the stratification of melanoma patients that may benefit from anti-PD-1 treatment.

## Figures and Tables

**Figure 1 cancers-13-02325-f001:**
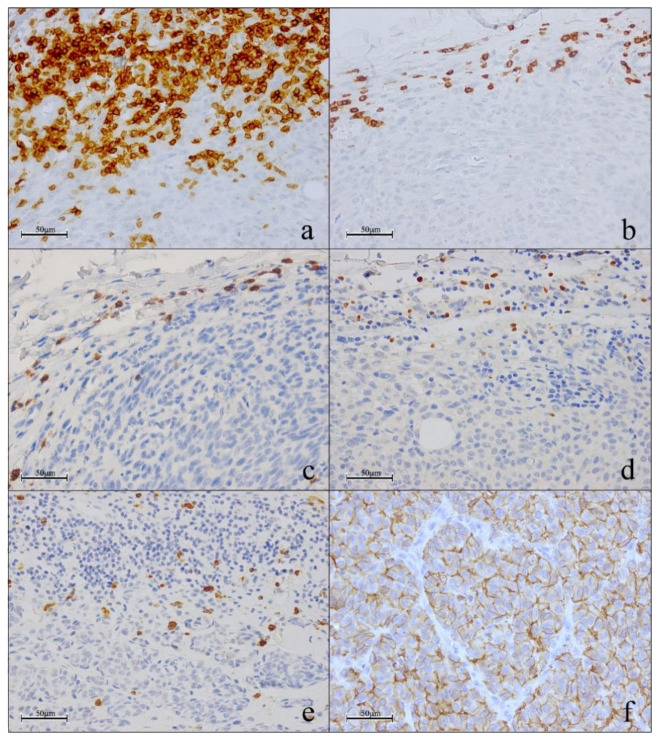
Single immunohistochemistry assay on peritumoral (**a**–**e**) or intratumoral (**f**) area of melanoma samples. Representative image of single immunostaining: expressions of high CD3 (**a**); low CD3 (**b**); CD8 (**c**); FOXP3 (**d**); GRZB (**e**); PD-L1 (**f**).

**Figure 2 cancers-13-02325-f002:**
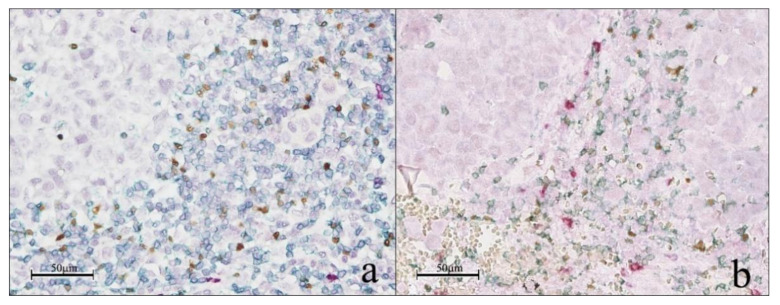
Multiplex immunohistochemistry. Representative image of multiplex immunohistochemistry with CD3 (green), GRZB (red) and FOXP3 (brown) for sample with low ratio (<−0.05) (**a**) and sample with high ratio (≥−0.05) (**b**).

**Figure 3 cancers-13-02325-f003:**
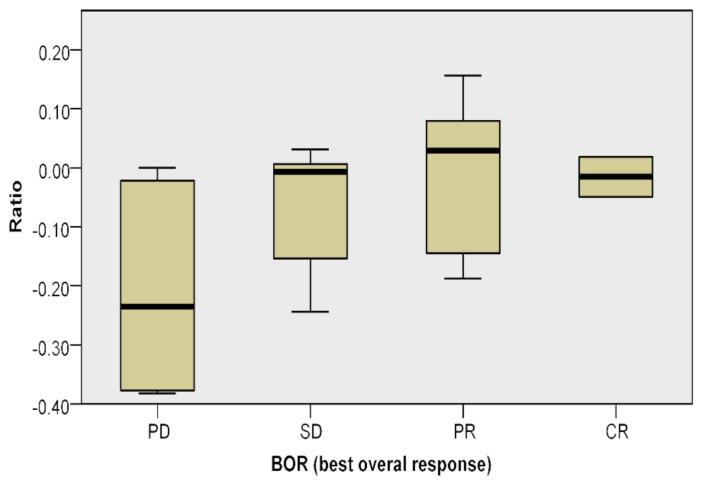
Kruskal–Wallis test. Relationship between the applied ratio and BORR.

**Figure 4 cancers-13-02325-f004:**
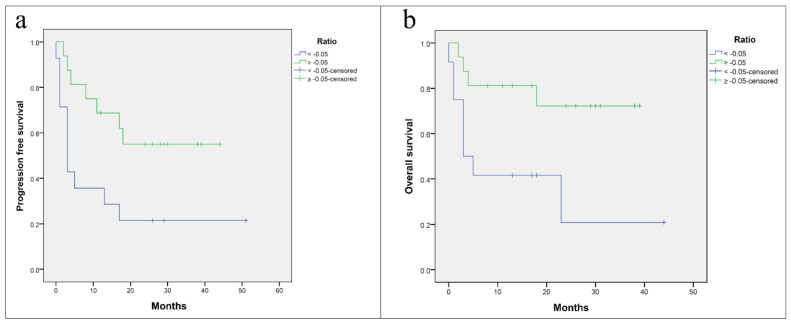
Kaplan–Meier curves. Correlation between progression-free survival (**a**) and overall survival (**b**) with ratio (cut-off: −0.05).

**Table 1 cancers-13-02325-t001:** Clinical–pathological features of melanoma patients. Thirty melanoma patients treated with nivolumab and their clinical–pathological features.

Characteristic		No. (%)
Age, years	≤67	15 (50)
	>67	15 (50)
Gender	Female	14 (46.7)
	Male	16 (53.3)
Disease stage	IIIC	1 (3.3)
	IV	29 (96.7)
M Category	M0	1 (3.3)
	M1A	8 (26.7)
	M1B	2 (6.6)
	M1C	14 (46.7)
	M1D	5 (16.7)
Primary vs metastases	PC	14 (46.7)
	MC	16 (53.3)
BRAF status	Wild-type	21 (70)
	Mutant	9 (30)
Line of treatment	1°	12 (40)
	2°	12 * (40)
	3°	5 ** (16.7)
	4°	1 *** (3.3)
Lactate dehydrogenase	Normal	21 (70)
	Elevated	6 (20)
	NA	3 (10)
Response	Complete response	2 (6.6)
	Partial response	8 (26.7)
	Stable disease	9 (30)
	Progressive disease	11(36.7)

* 11 out of 12 patients received ipilimumab as first line, 1 out of 12 vemurafenib; ** all 5 patients received ipilimumab as first line and vemurafenib as second line; *** this patient was enrolled in a clinical trial as first line; then, he received ipilimumab as second line and vemurafenib as third line.

**Table 2 cancers-13-02325-t002:** Tumor pathologic characteristics of primary melanomas. The considered features of primary melanomas (*n* = 14) were tumor size, ulceration and mitoses.

Characteristic		No.	(%)
Ulceration	Absent	4	(28.6)
	Present	10	(71.4)
Mitoses	<1/mm^2^	2	(14.3)
	>1/mm^2^	12	(85.7)
Tumor size	pT1	2	(14.3)
	pT2	2	(14.3)
	pT3	6	(42.9)
	pT4	4	(28.6)

**Table 3 cancers-13-02325-t003:** Relationship among single TIL biomarkers with the pathological features. Representation of the intersection between the variables using the mean ranks of the analyzed groups, obtained with the Mann–Whitney test and Kruskal–Wallis rank test.

Biomarker	Gender			Tumor			BRAF			Best Overall Response Rate				
	F (*n* = 14)	M (*n* = 16)	*p*-Value	Primary (*n* = 14)	Metastatic (*n* = 16)	*p*-Value	V600 (*n* = 9)	WT (*n* = 21)	*p*-Value	PD (*n* = 11)	SD (*n* = 9)	PR (*n* = 8)	CR (*n* = 2)	*p*-Value
CD3	17.79	13.5	0.193	15.75	15.28	0.886	13.5	16.36	0.422	14.27	19.24	13.38	10.75	0.313
CD8	16.64	14.5	0.525	15.18	15.78	0.854	14.17	16.07	0.594	17.09	16.06	13.25	13.25	0.79
FOXP3	16.93	14.25	0.423	16.75	14.41	0.473	17.67	14.57	0.397	18.18	15.33	12.56	13.75	0.57
GRZB	16.14	14.94	0.728	16.43	14.69	0.608	15.72	15.4	0.929	12.86	14.5	19.44	18.75	0.389
PD-L1	13.25	17.47	0.193	13.61	17.16	0.275	15.77	15.43	0.695	14.45	15.67	14.38	25	0.292

No statistically significant differences were shown between the number of single biomarkers (CD3^+^, CD8^+^, FOXP3, GRZB and PD-L1) and clinicopathological features.

**Table 4 cancers-13-02325-t004:** Spearman’s rank correlation coefficient. Correlations between the expressions of single markers with each other.

Biomarker		CD3	CD8	FOXP3	GRZB
CD8	Correlation	0.785			
	*p*-value	**<0.001**			
FOXP3	Correlation	0.315	0.547		
	*p*-value	0.09	**0.002**		
GRZB	Correlation	0.470	0.380	0.157	
	*p*-value	**0.009**	**0.038**	0.407	
PD-L1	Correlation	0.057	0.184	0.106	0.214
	*p*-value	0.764	0.329	0.576	0.256

In bold are the significant *p*-values.

**Table 5 cancers-13-02325-t005:** Relationship between ratio and pathological features. Representation of the intersection among the ratio and some patients’ clinical characteristics, using the mean ranks of the analyzed groups, obtained with the Mann–Whitney test and Kruskal–Wallis rank test.

Biomarker	Gender			Tumor			BRAF			Best Overall Response Rate				
	F (*n* = 14)	M (*n* = 16)	*p*-Value	Primary (*n* = 14)	Metastatic (*n* = 16)	*p*-Value	V600 (*n* = 9)	WT (*n* = 21)	*p*-Value	PD (*n* = 11)	SD (*n* = 9)	PR (*n* = 8)	CR (*n* = 2)	*p*-Value
Ratio	16.21	14.88	0.697	13.43	17.31	0.24	13.67	16.29	0.476	9.55	16.56	21.25	20.5	0.025

## Data Availability

The datasets used and/or analyzed during the current study are available from the corresponding author on reasonable request.

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
