# Peer review of "The Ratio of GrzB+ − FoxP3+ over CD3+ T Cells as a Potential Predictor of Response to Nivolumab in Patients with Metastatic Melanoma"

_cancers, 2021, doi:10.3390/cancers13102325_

Round 1

Reviewer 1 Report

The authors have presented a nicely written manuscript that attempts to stratify patients who will respond and those will not respond to treatment with PD-1 based immune checkpoint blockade, based on an applied ratio of the difference between GRZB (as a marker of cytotoxic T-cell function) and FOXP3 (as a marker for immunosuppressive T regulatory cells) cells to total CD3+ cells.

The study offers some intriguing points, and points to the potential for quick immunohistochemistry testing analysis of tumor specimens to help predict response to treatment.  While it does generate some important hypotheses, it does need to be further explored and validated.  Of note, the cohort included a total of 30 melanoma samples, from patients treated with PD-1 blockade.  Furthermore, the applied ratio cutoff of -0.05 was made based on the available data set, and may not necessarily be generalizable across larger data sets without further confirmation studies.  Nevertheless, it provides a readily applied methodology to be considered as it could be implemented at most medical centers.

We humbly submit the following comments, which we hope may strengthen the manuscript and increase its value to the reader audience:

-Could the authors comment as to why they used both primary and also in-transit metastatic samples?  Why not metastases from other anatomic sites?  On the one hand, one may argue that they are both located within the cutaneous tissues and thus similar; on the other hand, one could also argue that a metastatic tumor may present a different tumor immune microenvironment altogether.  The authors found that PD-L1 expression seemed higher (43.7%) in the in-transit metastases, compared to primary melanomas (21.4%).  This may warrant attention in the discussion.

-Could the authors comment on how prior lines of therapy might affect the presence of immune cells in these samples?  We know that other available therapies, such as ipilimumab, and also BRAF targeted agents may affect the immune response in melanoma.  Consider including this in the discussion.

-While GRZB and FOXP3 certainly may be nice indicators of either a cytotoxic or immunosuppressive TIME, the authors should also consider/comment on whether other cell types would be expected to also affect immune responses.

Minor edits/grammatical errors:

Simple summary:
-on line 19, would replace the word “to” with “on” in the phrase “based to the numerical difference”.
-on line 21, would change the word “patient” to be “patients”.

Introduction:
-line 66: the phrase “Granzyme B (GRZB) is the most abundant …” seems incomplete.  Perhaps it should state “Granzyme B (GRZB) is the most abundant of these”
-line 71: consider adding the word “most” before “prevalent”
-line 72-73: the sentence ending with “recent studies revealed that …. correlates with melanoma prognosis” is a bit confusing.  Please consider rephrasing this.  May want to simply say, “other types of immune cells, such as M1 or M2 macrophages, also correlate with melanoma prognosis”.

Table 3 and 5:
-consider using decimal points, instead of commas, for p-values (i.e. “0.193” instead of 0,193”)

Discussion:
-line 265-266: I believe there is an error with the phrase “especially in compared with ipilimumab”.  Consider revising as “especially in combination with ipilimumab, which leads to a significant, greater clinical benefit compared to anti-PD-1 alone.”
-line 271: should say “unleashes” rather than “unleash”
-line 274: consider using the word “compose” or “comprise”, rather than “create”
-line 275: remove the word “the” before “dampening or enhancing”
-line 288: should be “markers” instead of “marker”
-lines 309-311:  Please check this sentence: it is a bit wordy, and confusing.  You may want to simply state that “ipilimumab increased the frequency of CD8+ and CD4+ T-cells but not FOXP3+ Tregs.”

Author Response

Dear Reviewer,

We would like to thank you for the attention in the critical review of the manuscript and for the requested revisions. In these regards, here a point-by-point response with an explanation of any request.

Point 1: Could the authors comment as to why they used both primary and also in-transit metastatic samples? Why not metastases from other anatomic sites? On the one hand, one may argue that they are both located within the cutaneous tissues and thus similar; on the other hand, one could also argue that a metastatic tumor may present a different tumor immune microenvironment altogether. The authors found that PD-L1 expression seemed higher (43.7%) in the in-transit metastases, compared to primary melanomas (21.4%). This may warrant attention in the discussion.
Response 1: According to your suggestion, we added comments regarding samples and PD-L1 expression, as follow:
• “both located in cutaneous tissues. For this reason, we decided to exclude other types of metastasis, such as lymph node and visceral ones, which certainly have a very different tumor microenvironment. Moreover, this choice was also related to the readily availability of these pre-treatment samples.” (Page 12 lines 386-389, in the Marked revised manuscript).
“Furthermore, limited data exist comparing PD-L1 status in primary versus metastatic melanoma lesions, with few matched sets available from the same patient. FFPE primary melanoma samples may be most readily available, but they may not reflect the overall immunologic state that currently exists in a given patient and may not capture the beneficial effect therapy is having on other sites of disease that are dependent on PD-L1 signaling.”. (Page11 lines 339-344, in the Marked revised manuscript).

Point 2: Could the authors comment on how prior lines of therapy might affect the presence of immune cells in these samples?
Response 2: According to your observations, we added a paragraph in discussion on how prior lines of therapy might affect the presence of immune cells in these samples and their functionality adding 2 references, as follow:
"Among all available patient samples, we selected exclusively the one excised in proximity to the start of nivolumab therapy. All samples were excised before the patient started nivolumab therapy. 12 patients were treated in the first line with nivolumab, while 18 patients were treated in the second, third, or even 4 line with it. We know that both immunotherapy and BRAF targeted agents may affect the presence of immune cells and their functionality [24, 25]." (Page 11 lines 320-325, in the Marked revised manuscript).

Point 3: While GRZB and FOXP3 certainly may be nice indicators of either a cytotoxic or immunosuppressive TIME, the authors should also consider/comment on whether other cell types would be expected to also affect immune responses.

Response 3: According to your suggestion, we added a paragraph in discussion on whether other cell types would be expected to also affect immune responses, as follow:
"We wanted to consider the key players of adaptive immune responses based on T cells because the activation and expansion of these cell types is necessary for long-term melanoma remission and for response to treatment. Furthermore, the intratumoral infiltration of activated lymphocytes derives from the other immune cells yet residing in the tumor niche, such as macrophages, neutrophils and dendritic cells that phagocytize dead melanoma cells and present cancer antigens activating secondary adaptive immune responses." (Page 11 lines 313-319, in the Marked revised manuscript).

Point 4: Minor edits/grammatical errors.
Response 4: According to your suggestion, we edit manuscript as follow:
- "based on the numerical difference" (Page 1 line 21, in the Marked revised manuscript).
- “patients” (Page 1 line 21, in the Marked revised manuscript)
- “Granzyme B (GRZB) is the most abundant of these” (Page 2 line 74, in the Marked revised manuscript)
- “most prevalent” (Page 2 line 79, in the Marked revised manuscript)
- “recent studies revealed that other types of immune cells, such as M1 or M2 macrophages, also correlate with melanoma prognosis (Page 2 lines 80-82, in the Marked revised manuscript)
- We used decimal points in Table 3 and 5 (Page 8 and page 10, in the Marked revised manuscript)
- “Some clinical trials have evaluated the efficacy of anti-PD1 treatments in melanoma, especially in combination with ipilimumab, which leads to a significant, greater clinical benefit compared to anti-PD-1 alone [20].” (Page 10 lines 288-292, in the Marked revised manuscript)
- “unleashes” (Page 11 line 297, in the Marked revised manuscript)
- “compose” (Page 11 line 300, in the Marked revised manuscript)
- “role in dampening or enhancing” (Page 11 line 301, in the Marked revised manuscript)
- “markers” (Page 11 line 326, in the Marked revised manuscript)
- “ipilimumab increased the frequency of CD8+ and CD4+ T-cells but not FOXP3+ Tregs.” (Page 12 lines 368-371, in the Marked revised manuscript)

Again, thank you so much, and please do not hesitate to contact me if you have other queries or concerns.

Sincerely,
Mariaelena Capone

Reviewer 2 Report

SCOPE: The article entitled “The ratio of GrzB+ − FoxP3+ over CD3+ T cells predicts response to nivolumab in patients with metastatic melanoma” does fulfill the scope of Cancers: basic and clinical studies on all tumor types.

Summary of the paper:

The article entitled “The ratio of GrzB+ − FoxP3+ over CD3+ T cells predicts response to nivolumab in patients with metastatic melanoma” by Scognamiglio and colleagues presents an application study in which 30 melanoma patient samples including primary and unmatched in-transit metastatic melanoma were analyzed in its peritumoral area for immune recruitment and composition. They stained the samples for the routine marker CD3, CD8, GrzB, FoxP3, MMR panel and PD-L1. They claim that a positive ratio of GRZB-FoxP3/CD3 predicts therapeutic response to nivolumab and correlates with a better progression free (PFS) and overall survival (OS).

Overall comment:

In general, the concept of the paper is well structured and clinical data are described in detail. Although the marker used in the manuscript have been described hundreds of time regarding response to checkpoint inhibitor therapy, the authors identified a new combination to ensure treatment response in advance.

Minor comments:

  1. CD3+ cells are not only T cells as claimed. NK cells also express CD3. The authors should perform a double positive staining to separate T cells from the NK cell population. For example they can use CD3+CD56+ double detection as described in the paper by Sharma et al. (DOI: 10.1158/1078-0432.CCR-18-0762) to identify the NK cell population in their cohort.
  2. In the results part, the authors claim that there is a direct correlation of CD3+CD8+FoxP3+GrzB+ Would be also interesting to see the ratio of GrzB-FoxP3/CD8 identifying CD8+Foxp3+ Treg cells.
  3. Which method was used for BRAF mutation detection? MiSeq?

Novelty of the publication:

The novelty of this publication is the possibility to use the ratio as a parameter for therapeutical decisions. As mentioned by the authors, several publications have shown IHC and IF of several markers (for example https://www.nature.com/articles/s41598-018-28944-3) but the calculation of the mentioned ratio based on IHC could present a simple and effective method to get first implications into patients response to the selected therapy.

Author Response

Dear Reviewer,

We would like to thank you for the attention in the critical review of the manuscript and for the requested revisions. In these regards, here a point-by-point response with an explanation of any request.

Point 1: CD3+ cells are not only T cells as claimed. NK cells also express CD3. The authors should perform a double positive staining to separate T cells from the NK cell population. For example they can use CD3+CD56+ double detection as described in the paper by Sharma et al. (DOI: 10.1158/1078-0432.CCR-18-0762) to identify the NK cell population in their cohort.
Response 1: Thank you for your valuable suggestions. Indeed, your comment is right, and we will consider to do this in a larger and more homogeneous sample cohort study.

Point 2: In the results part, the authors claim that there is a direct correlation of CD3+CD8+FoxP3+GrzB+ Would be also interesting to see the ratio of GrzB-FoxP3/CD8 identifying CD8+Foxp3+ Treg cells.
Response 2: Thank you for your valuable suggestions. In the IHC multistaining protocol the CD8 has not been evaluated so at this moment we cannot evaluate the colocalization of FOXP3 and CD8. Indeed, your comment is right, and we will consider your observation for a larger and more homogeneous sample cohort study.

Point 3: Which method was used for BRAF mutation detection?
Response 3: Thank you for your question. The BRAF mutational status was determined by Real-Time PCR.

Again, thank you so much, and please do not hesitate to contact me if you have other queries or concerns.
Sincerely,
Mariaelena Capone

Reviewer 3 Report

Anti-PD-1 treatment prolonged overall survival in patients with advanced melanoma; however, a significant number of patients do not respond to treatment and, currently, there are no clinically available biomarkers to predict responding patients in melanoma. The identification of predictive biomarkers is therefore a hot topic and a largely unmet clinical need.

The Authors used a multiplex immunostaining approach to evaluate the 
immune cell infiltration in both primary and unmatched in-transit metastatic melanoma lesions of patients retrieved before treatment with anti-PD-1. They evaluated the presence of immunological markers in both the intratumoral and peritumoral areas, proposing a lymphocytes ratio that could be useful for treatment personalization.

The results are well presented, the conclusions are supported by the results and main limitations of the study are clearly stated in the discussion section; I would only add to the limitations of the study the heterogeneity of the study population, in particular regarding to line of treatment and type of biopsy (primary tumor vs in-transit metastasis). I also suggest a slight change of the wording of the title to properly reflect the potential predictive role of the proposed biomarker.

Author Response

Dear Reviewer,

We would like to thank you for the attention in the critical review of the manuscript and for the requested revisions. In these regards, here a point-by-point response with an explanation of any request.

Point 1: I would only add to the limitations of the study the heterogeneity of the study population, in particular regarding to line of treatment and type of biopsy (primary tumor vs in-transit metastasis).
Response 1: Thank you for your valuable suggestions. We have added in the discussion some comments regarding these issues, as follow:
• “both located in cutaneous tissues. For this reason, we decided to exclude other types of metastasis, such as lymph node and visceral ones, which certainly have a very different tumor microenvironment. Moreover, this choice was also related to the readily availability of these pre-treatment samples.” (Page 12 lines 386-389, in the Marked revised manuscript).
• "Among all available patient samples, we selected exclusively the one excised in proximity to the start of nivolumab therapy. All samples were excised before the patient started nivolumab therapy. 12 patients were treated in the first line with nivolumab, while 18 patients were treated in the second, third, or even 4 line with it. We know that both immunotherapy and BRAF targeted agents may affect the presence of immune cells and their functionality [24, 25]." (Page 11 lines 320-325, in the Marked revised manuscript).

Point 2: I also suggest a slight change of the wording of the title to properly reflect the potential predictive role of the proposed biomarker.
Response 2: Thank you for your valuable suggestion and according to the same comment of Editor, we changed the Title as follow: “The ratio of GrzB+ − FoxP3+ over CD3+ T cells as a potential predictor of response to nivolumab in patients with metastatic melanoma”.

Again, thank you so much, and please do not hesitate to contact me if you have other queries or concerns.

Sincerely,
Mariaelena Capone

Reviewer 4 Report

In this paper, the Authors present an analysis of different immune cell infiltrate biomarker, as a predictive and prognostic tool for patients with melanoma receiving nivolumab. The work is interesting, however its presentation should be improved.

Abstract:

Overall, the abstract is confusing and should be rewritten.

Lines 26-27: the Authors state that “Physical interactions between immune and tumor cells within the TME is essential for effective antitumor immunotherapy suggesting the need to define an immune score model which can help to predict an efficient immunotherapeutic response”. Please mitigate this comment, since the cited mechanism acts together with many other and should therefore be listed as a part of features which finally contribute in determining response to treatment.

Line 29: what do the Authors mean with “unmatched”?

Introduction:

I suggest to provide a more precise background on the available treatment strategies for cutaneous melanoma.

Lines 48-49: These approaches have revolutionized the treatment of melanoma, but are not effective in all 49 patients, resulting in responder and non-responder populations. This sentence is true but, again, the differences among responders and non-responders are more complex. I suggest the Authors not to dichotomize, but to discuss the elements contributing to response and resistance to immunotherapy.

I suggest, as an example, to cite “Hanahan D, Weinberg RA. Hallmarks of cancer: the next generation. Cell. 2011 Mar 4;144(5):646-74. doi: 10.1016/j.cell.2011.02.013. PMID: 21376230”.

In the Introduction, the Authors should provide a more detailed list of factors contributing to response and resistance to treatment. This paragraph contains a lot of relevant information on the background of the research and should be explained in a clearer way.

Materials and Methods:

I would suggest the Authors to provide details on the baseline characteristics of the study population before talking about tumor samples.

As an example: We analyzed tumor samples of xx patients with metastatic melanoma treated with nivolumab from … to … at our Institution.

Lines 97-98: please explain what does “unmatched” mean in this specific context.

Results:

Please provide data on the median follow up of the study population.

Why have patients with BRAF mutant disease been treated with anti-PD1 as first line therapy? This indeed is a strange choice, or at least it should be explained since this can be motivated by a more indolent or less extended disease, thus introducing a potential bias of the study.

Please correct the Title of Table 2, since there is a replicated sentence.

Did the Authors find any differences in the disposition of CD8+, CD3+, FOXP3, and GRZB+ cells? It is well recognized that the presence of immune infiltrate is important, along with its disposition in “strategic areas”. Please elaborate.

Table 3 Title. This sentence should be moved to the caption of the table: “Intersections between the variables represents the mean ranks of the analyzed groups, obtained with the Mann-Whitney test and Kruskal–Wallis rank test”.

The same for Table 5 title.

Discussion:

Overall, the discussion should be shortened as it contains a lot of repetition.

Among the limitations of the study it should be discussed the length of PFS which is quite short. Moreover, no data on the median follow up of the population nor on the subsequent treatment received at the time of disease progression have been provided. This can impact on OS.

Line 259: the role of the immune system in the pathogenesis of melanoma does not depend from the immunologic origin of the disease. Please correct and rephrase.

Line 263: Please consider to remove this sentence to avoid repetitions, as this has already been discussed “These therapies have dramatically improved the prognosis of patients with metastatic melanoma”.

Lines 264-266: please rephrase: anti-PD1 antibodies have demonstrated to provide better survival outcomes with more manageable safety profiles, as compared with ipilimumab and has become the standard first line immunotherapy. Anyway, the reference provided is wrong (20) as it refers to the clinical trial of combined nivo + ipi. CheckMate 066 and 067 should be cited, along with the results of the KEYNOTE-006 trial.

Lines 284-287 should be removed since this data has been provided in the Materials and Methods section and shouldn’t be repeated in the discussion.

Conclusions section should be more general and introduce some open questions, along with current research in the field. Please avoid repeating the results of the paper.

Author Response

Dear Reviewer,

We would like to thank you for the attention in the critical review of the manuscript and for the requested revisions. In these regards, here a point-by-point response with an explanation of any request.

Point 1: Lines 26-27: the Authors state that “Physical interactions between immune and tumor cells within the TME is essential for effective antitumor immunotherapy suggesting the need to define an immune score model which can help to predict an efficient immunotherapeutic response”. Please mitigate this comment, since the cited mechanism acts together with many other and should therefore be listed as a part of features which finally contribute in determining response to treatment.
Response 1: We have added a short list of the other components of the TME, equally important in contributing an immunotherapeutic response. (Page 1 Lines 28-30 in the Marked revised manuscript)

Point 2: What do the Authors mean with “unmatched”?
Response 2: The word “unmatched” indicates that primary and metastatic melanoma lesions were not from the same patient. We also explained it in the text as follow: “(Page 3 Lines 111-112 in the Marked revised manuscript)

Point 3: I suggest to provide a more precise background on the available treatment strategies for cutaneous melanoma. Lines 48-49: These approaches have revolutionized the treatment of melanoma, but are not effective in all patients, resulting in responder and non-responder populations. This sentence is true but, again, the differences among responders and non-responders are more complex. I suggest the Authors not to dichotomize, but to discuss the elements contributing to response and resistance to immunotherapy. I suggest, as an example, to cite “Hanahan D, Weinberg RA. Hallmarks of cancer: the next generation. Cell. 2011 Mar 4;144(5):646-74. doi: 10.1016/j.cell.2011.02.013. PMID: 21376230”. In the Introduction, the Authors should provide a more detailed list of factors contributing to response and resistance to treatment. This paragraph contains a lot of relevant information on the background of the research and should be explained in a clearer way.
Response 3: Thank you for your correct observations, but the aim of the work is not focused on the mechanisms of response or resistance to anti-PD-1 treatment, but on identification of new biomarkers capable of predicting the clinical benefit to treatment. However, minor modifications of the text have been made.

Point 4: I would suggest the Authors to provide details on the baseline characteristics of the study population before talking about tumor samples and to explain what does “unmatched” mean in this specific context.
Response 4: According to your observations, we added a paragraph in the section of Materials and Methods- 2.1. Tumor samples as follow: “From the biobank of the Istituto Nazionale Tumori Fondazione “G. Pascale” (Naples, Italy). we recovered excised melanoma tumor samples from thirty patients (16 males, 14 females, median age 66,5 years) subsequently treated with nivolumab as monotherapy from September 2014 to November 2017. Nivolumab was administrated at the dosage of 3 mg/kg every 2 weeks until disease progression or unacceptable toxicity appeared. In particular, we retrospectively recovered 14 primary melanomas (46.7%) and 16 unmatched in-transit melanoma metastases (53.3%), i.e. the primary and melanoma metastases were not from the same patient. All tumor samples were excised from melanoma patients before treatment with anti-PD-1.” (Page 3 lines 105-116, in the Marked revised manuscript).
Also, we added a short section below the Table 1 indicating which treatment patients received prior nivolumab treatment.

Point 5: Please provide data on the median follow up of the study population. Thank you for your suggestion.
Response 5: We added a sentence as follow: “Median OS for these patients was of 20 months and median PFS was 16 months; median time of follow-up for them was of 20 months (range 1-54 months).” (Page 5-6 Lines 187-189, in the Marked revised manuscript).

Point 6: Why have patients with BRAF mutant disease been treated with anti-PD1 as first line therapy? This indeed is a strange choice, or at least it should be explained since this can be motivated by a more indolent or less extended disease, thus introducing a potential bias of the study.
Response 6: Decisions about immunotherapy, targeted therapy, or the combination of immunotherapy with targeted therapy require an oncologist to evaluate multiple factors to select which treatment option is best for the patient. The clinician's choice of first-line treatment for patients with BRAF mutation was made in accordance with International Guidelines and based on the patient’s tumor burden and to the availability of the result of BRAF status.

Point 7: Please correct the Title of Table 2, since there is a replicated sentence.
Response 7: We have changed the caption. (Page 6 lines 194-196, in the Marked revised manuscript).

Point 8: Did the Authors find any differences in the disposition of CD8+, CD3+, FOXP3, and GRZB+ cells? It is well recognized that the presence of immune infiltrate is important, along with its disposition in “strategic areas”. Please elaborate.
Response 8: We evaluated both intratumoral and peritumoral areas, but in the intratumoral regions the numerical evaluation of markers was very low and no statistically significant, so we focused on the peritumoral region of all cases.

Point 9: Table 3 and Table 5 title.
Response 9: According to your suggestion, we have changed the text of Table 3 as follow: “Table 3. Relationship among single TILs biomarkers with the pathological features. Representation of the intersection between the variables using the mean ranks of the analyzed groups, obtained with the Mann-Whitney test and Kruskal – Wallis rank test.”, Page 7 lines 219-223, in the Marked revised manuscript), and the text of Table 5 as follow: “Representation of the intersection among the ratio and some patients clinical characteristics, using the mean ranks of the analyzed groups, obtained with the Mann-Whitney test and Kruskal – Wallis rank test. (Page 9 lines 260-264, in the Marked revised manuscript).

Point 10: Overall, the discussion should be shortened as it contains a lot of repetition. Among the limitations of the study it should be discussed the length of PFS which is quite short. Moreover, no data on the median follow up of the population nor on the subsequent treatment received at the time of disease progression have been provided. This can impact on OS.
Response 10: Thank you for these comments, in particular because you have highlighted a typo in the text and we have corrected and added the medians of OS, PFS and follow-up in the Results section (Page 5-6 Lines 187-189, in the Marked revised manuscript). Furthermore, we followed the patients even after the end of treatment with anti-PD-1 but due to the small number of patients we hadn’t evaluated the impact of the subsequent treatment on OS.

Point 11: Line 259: the role of the immune system in the pathogenesis of melanoma does not depend from the immunologic origin of the disease. Please correct and rephrase.
Response 11: According to your comment, we changed the sentence as follow: “Based on the knowledge of the melanoma biology and its immunogenicity, new therapeutic strategies have been developed with antibodies directed to specific targets, including anti-PD-1 and anti-CTLA-4 [18,19].” (Page 10 lines 282-288, in the Marked revised manuscript).

Point 12: Line 263: Please consider to remove this sentence to avoid repetitions, as this has already been discussed “These therapies have dramatically improved the prognosis of patients with metastatic melanoma”.

Response 12: We have rightly eliminated this sentence. (Page 10 lines 286-287, in the Marked revised manuscript).

Point 13: Lines 264-266: please rephrase: anti-PD1 antibodies have demonstrated to provide better survival outcomes with more manageable safety profiles, as compared with ipilimumab and has become the standard first line immunotherapy. Anyway, the reference provided is wrong (20) as it refers to the clinical trial of
combined nivo + ipi. CheckMate 066 and 067 should be cited, along with the results of the KEYNOTE-006 trial.
Response 13: Thank you for your suggestions. We have changed the sentence and the reference as follow: “Some clinical trials have evaluated the efficacy of anti-PD1 treatments in melanoma, especially in combination with ipilimumab, which leads to a significant, greater, clinical benefit compared to anti-PD-1 alone [20].” (Page 10 lines 287-292 in the Marked revised manuscript).

Point 14: Lines 284-287 should be removed since this data has been provided in the Materials and Methods section and shouldn’t be repeated in the discussion.
Response 14: We have changed the sentence. (Page 11 lines 309-313, in the Marked revised manuscript).

Point 15: Conclusions section should be more general and introduce some open questions, along with current research in the field. Please avoid repeating the results of the paper.
Response 15: We have changed the paragraph following your instructions and those of the journal, as follow: “In this study, for the first time, we have introduced a new Ratio (Ratio=# of GRZB – # of FOXP3)/# of CD3) to evaluate if the peritumoral regions were more cytotoxic or immune suppressive and if it could have had an impact on the response to therapy and its duration. Although the number of samples analyzed is small, we found that higher ratio positively correlated with BORR and with the PFS and OS, indicating that it could be useful in the stratification of melanoma patients that may benefit from anti-PD-1 treatment.” (Page 13 lines 402-409, in the Marked revised manuscript).

Again, thank you so much, and please do not hesitate to contact me if you have other queries or concerns.

Sincerely,
Mariaelena Capone

Round 2

Reviewer 1 Report

Well done with this paper.  Agree with the responses and corresponding edits; the paper looks much improved in the current format.

Author Response

Thank you

Reviewer 2 Report

All open questions are now addressed.

Author Response

thank you

Reviewer 3 Report

The authors addressed all my concerns. I have no further comments.

Author Response

thank you